# Mechanical work derived using markerless motion capture provides a valid indication of acute neuromuscular fatigue in tennis

Julie Emmerson[1,2], Laurie Needham[1,2], Sean Williams[1,3], Steffi L. Colyer[1,2]*

1 Department for Health, University of Bath, Bath, United Kingdom, 2 Centre for the Analysis of Motion, Entertainment Research and Applications, University of Bath, Bath, United Kingdom, 3 Centre for Health and Injury and Illness Prevention in Sport, University of Bath, Bath, United Kingdom

* s.colyer@bath.ac.uk

## Abstract

Whilst workload monitoring is key to managing performance and injury risk, effective strategies in tennis are not well-established. Mechanical work (external and internal components), measurable through markerless motion capture approaches, provides a novel way to quantify workload. To determine the suitability of this approach, this study investigated the association between mechanical work and acute neuromuscular fatigue. Fifteen tennis players completed a tennis-specific on-court fatiguing protocol interspersed with sprint tests. Peak sprint velocity (measure of acute neuromuscular fatigue) was correlated against mechanical work for each player. Pooled correlations (accounting for inter-individual differences) for total (external + internal) and external work, as well as external work estimated from centre of mass (CoM) proxies (pelvis and bounding box methods) were comparable, with values of −0.93 to −0.92. Although the calculated work done varied greatly between methods (~40% for pelvis method), these differences were systematic. All methods therefore appeared to provide an indication of individual players' workload (acute neuromuscular fatigue), despite the absolute values of the CoM proxy methods being inaccurate. Therefore, we recommend using the approximations only when monitoring an individual player in similar contexts. These findings present the markerless approach as a promising tool that could be implemented for non-invasive, on-court workload monitoring in tennis.

## Introduction

The incidence of injuries in tennis has been associated with both high [1] and sudden increases in [2] acute workload, with workload broadly defined here as a concept referring to the training dose experienced by an athlete. This can be problematic with the demanding tournament schedules often experienced in tennis. A significant increase in medical withdrawals beyond the fourth round of tournaments has

**Data availability statement:** All relevant data underlying this study are available from the Figshare repository (https://doi:10.6084/m9.figshare.30328375).

**Funding:** This research was funded by EPSRC (Engineering and Physical Sciences Research Council), through CAMERA, the Centre for the Analysis of Motion, Entertainment Research and Applications (EP/M023281/1 and EP/T022523/1). The funders had no role in study design, data collection and analysis, decision to publish, or preparation of the manuscript.

**Competing interests:** The authors have declared that no competing interests exist.

previously been reported [3], demonstrating the increased risk associated with playing multiple matches with limited recovery. Monitoring workload can be a useful tool in planning and adjusting training and competition schedules. Differences in 'load' have previously been reported between training and competition [4,5] and between playing on different court surfaces [6]. This information can be invaluable to players' tournament preparations.

Although 'load' monitoring is common practice across many sports, there exists no established gold-standard metric. Commonly used metrics are either unvalidated or validated against other unvalidated measures [7]. The multidimensional construct of 'workload' can be separated into two components: external and internal. External 'load' refers to the physical training carried out by the athlete, whilst internal 'load' refers to the athlete's psychophysiological response to the physical training [8]. Whilst monitoring both external and internal 'load' in parallel is important for understanding the dose-response nature of training, this study will focus on external mechanical 'load', which better explains the causal link between loading and injury [9].

In tennis, evidence for the utility of 'load' monitoring is sparse [10] and typically relies on simple measures of frequency and duration of training sessions [11] or hitting volume and stroke analysis, which can be obtained from wearable sensors [12–14]. There is a paucity of evidence quantifying player movement, utilising Hawk-Eye video tracking data [15] or wearable technology [16], but this is limited to distances covered in velocity zones with arbitrary thresholds, which does not give an overall picture of the mechanical demand on the player. Global (GPS) and Local (LPS) Positioning Systems can be used for tracking player movement, but there exists uncertainty over the accuracy and reliability of these approaches for rapid movements involving accelerations and changes of direction [17,18]. In addition, athletes may find wearable technology to be uncomfortable and intrusive to their technique, especially when worn for long durations [19]. Video-based tracking offers an alternative and non-invasive approach to wearable technology.

To date, video-based player tracking in tennis has predominantly been limited to traditional computer vision techniques based on 2D background subtraction and binary thresholding [20–23]. More recently, deep learning-based object detection methods have been used on 2D broadcast footage [24]. Using bounding boxes or silhouettes, these approaches typically either track the centroid or the estimated mid-feet position (based on the bottom edge of the bounding box or silhouette) on the court plane. Whilst tracking these points is computationally cheaper, and hence potentially has more utility, than tracking the body centre of mass (CoM), there are likely to be inconsistencies between movement of the CoM and these proxy points, especially when the player is in unusual poses. This will introduce small errors for each movement that will accumulate over time. However, the emergence of artificial intelligence-driven pose estimation algorithms has opened up the possibility for measuring full-body kinematics, and hence gaining more insight regarding player movement. This could allow for the measurement of mechanical work, which has recently gained interest as a potential metric for workload monitoring [25].

Mechanical work is a biomechanical metric that captures the high mechanical demands of cumulative accelerations and decelerations, which are not reflected in common distance- or velocity-based metrics. It can be split into external (work done to accelerate and raise CoM) and internal (work done to accelerate segments relative to CoM) components [26]. We have previously shown that mechanical work can be measured using a custom markerless motion capture system, with strong (external) and moderate (internal) agreement against a criterion marker-based motion capture system [27]. This could offer a new and non-invasive video-based approach to workload monitoring, with the potential to reach a far wider pool of players in the future than current 'load' monitoring technologies are capable of. However, the utility of this system as a 'load' monitoring tool has not yet been evaluated in an applied setting.

Assessing the validity of workload metrics is challenging due to the multidimensional nature of this construct, along with there being no established gold-standard measure [28]. Large-scale longitudinal monitoring of workload, performance and injury could help to establish relationships between these variables, although contributions and interactions of other factors, as well as individual variations, make this non-trivial [29]. Alternatively, it has been suggested that workload can be assessed through acute performance decrements [7]. Although the relationship between workload and fatigue is complex [30], the immediate physiological and mechanical strain of training could be inferred through a player's acute neuromuscular fatigue response, which can provide valuable insight into the stress imposed on the body. Therefore, the relationship between workload and acute neuromuscular fatigue could be used to indirectly assess the suitability of workload metrics.

The first aim of this study was to assess whether mechanical work measured with our custom markerless motion capture system is an appropriate measure of workload. This was addressed by associating the mechanical work done during a tennis-specific on-court fatiguing protocol with a measure of acute neuromuscular fatigue. The second aim was to determine if an approximation of mechanical work, that reduces data processing requirements, would be sufficient for this application. The approximations tested were neglecting the internal work and using a fixed point as a proxy for CoM.

## Materials and methods

### Participants

Fifteen tennis players (10 males [1.82 ± 0.06 m, 77.2 ± 4.7 kg], five females [1.71 ± 0.05 m, 67.2 ± 10.6 kg]) were recruited between 01/11/2023 and 17/04/2024 and provided written informed consent. The study was approved by the Research Ethics committee at the University of Bath (reference: EP22027). Players met the criteria for tier 2 (trained / developmental) or tier 3 (highly trained / national level) of the participant classification framework proposed by McKay and colleagues [31].

### Equipment and experimental protocol

Each player attended one data collection session on an indoor tennis court, during which they completed a tennis-specific fatiguing protocol (Fig 1). Prior to the protocol, players completed their own warm up, a practice set of the protocol and a baseline sprint test. A complete cycle of the protocol consisted of three sets and a maximum effort sprint. One set consisted of a serve (from the deuce side) followed by eight groundstrokes, alternating between forehand and backhand sides, with balls dropped into target zones at a set frequency given by a metronome to ensure consistency. The target zones were positioned 1.25 m forward from the baseline and 1 and 1.5 m in from the tramlines for males and females respectively. The frequency was set at one ball every 2 s and 2.5 s for males and females respectively. Between sets, a 20-s rest was given, which was reduced by 5 s after the fifth and tenth cycles to ensure the participant became fatigued within a practical time scale. The third set was immediately followed by a maximum effort sprint back-and-forth along the baseline and up the tramline to the net (Fig 2). One minute of rest was given after each cycle and this was repeated until the player reached volitional exhaustion. Reduction in peak forward CoM velocity (compared to the baseline) during each sprint to the net was taken as a valid measure of acute neuromuscular fatigue [32].

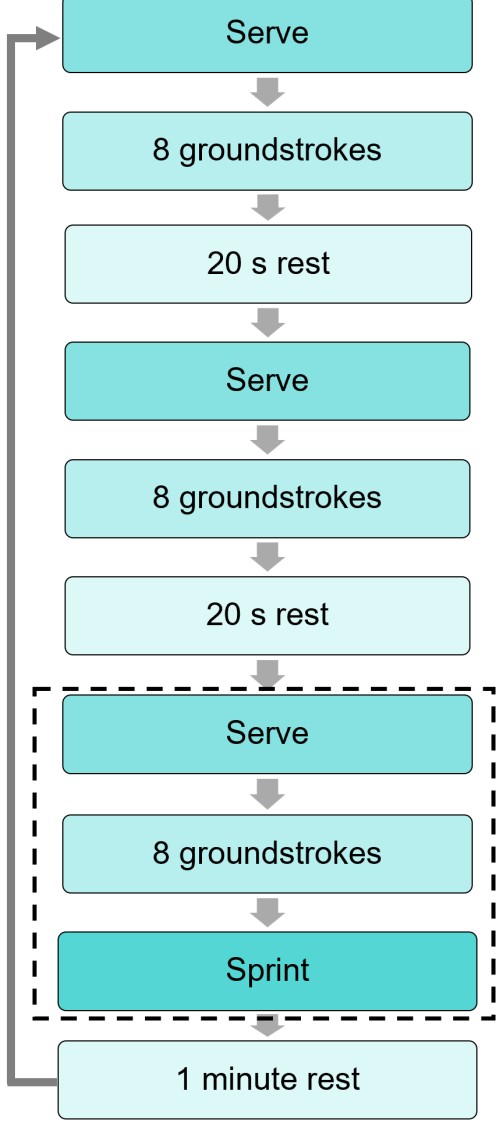

**Fig 1. Outline of the fatiguing protocol.** The protocol was repeated until the player reached volitional exhaustion. The 20-s rest was decreased by 5 s after the fifth and tenth cycles. The dashed box shows the section of each cycle that was captured by the markerless system.

A high-definition 8-camera system (JAI sp5000c, JAI ltd, Denmark), recording at 200 Hz, was used to capture the third set only of each cycle, including the sprint. This was due to excessive data saving times, with each capture lasting 30 s. As the task was identical across sets, the work done in each of the first two sets was assumed to be sufficiently similar to that of the captured third set. The cameras were positioned around the half-court being used, with as much of the half-court as possible in each field of view. The camera system was calibrated using observations of a binary dot matrix and a sparse bundle adjustment (as described in [33]). Additionally, players wore a heart rate monitor (Polar H10, Polar Electro Oy, Finland) for the duration of the protocol.

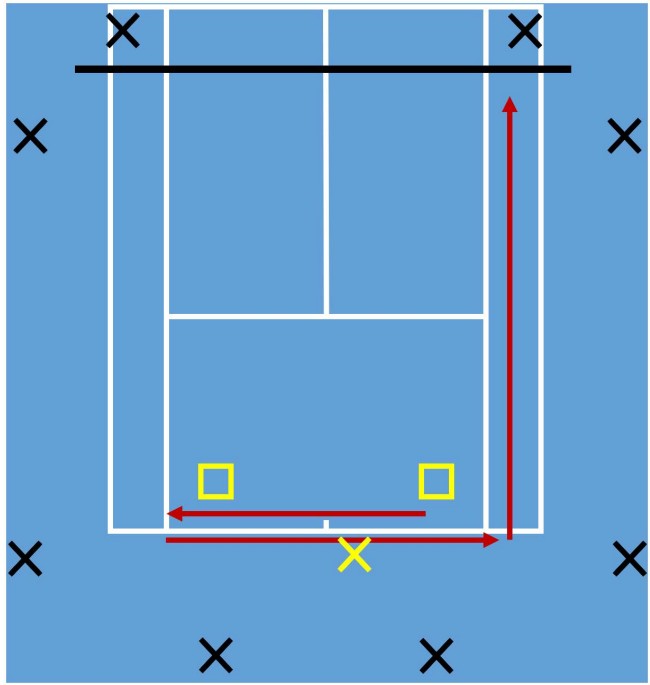

**Fig 2. Diagram of the experimental set-up on the tennis court.** Black crosses show approximate camera positions; the yellow cross shows the serving position; yellow squares show the ball target zones (player moves to the left after serving); red arrows show directions of the sprint (starting from the right target zone and finishing at the net).

## Markerless motion data processing

The processing of the markerless motion capture data followed the workflow presented by Needham and colleagues [33]. First, 2D sparse pose estimation was performed on every frame for each camera using a pre-trained model through MMPose [34]. The algorithms implemented were Faster R-CNN [35] for the initial human detection, followed by HRNet [36] with DARK plug-in [37], pre-trained on the COCO-Wholebody dataset [38]. An occupancy maps approach [33] was used to associate detections between viewpoints and across frames, assigning an identity to each track. The track corresponding to the player was identified based on the known approximate starting position of the player. The 2D keypoints were reconstructed in 3D by finding the least-squares solutions for the intersection of back-projected rays from each camera, using the appropriate calibrations and discarding any outliers. A bi-directional Kalman filter was then applied to the keypoint trajectories. These filtered trajectories were used to drive the motion of a constrained rigid body model in OpenSim [39]. The model (available at https://github.com/julieemmerson/opensim_tennis_model) is described in [27], with a virtual marker-set representing the keypoints used (COCO body keypoints, with additional foot and hand keypoints; Fig 3). Models were scaled for each subject using a static calibration trial, with segment mass and inertia properties based on the work of de Leva [40]. The OpenSim inverse kinematics (IK) tool was used to find a global optimisation of pose for each frame of experimental data. Resulting joint angles were filtered using a low-pass 4th order Butterworth filter with a 6 Hz cut-off frequency, determined by a residual analysis. Segment and CoM kinematics were exported for further analyses in Python (Python 3.10).

## Mechanical work computations

Mechanical work was calculated in line with the methods of Cavagna and Kaneko [26]. CoM kinetic (KE) and potential (PE) energies were calculated and summed to give total CoM mechanical energy at every timepoint across the 30-s

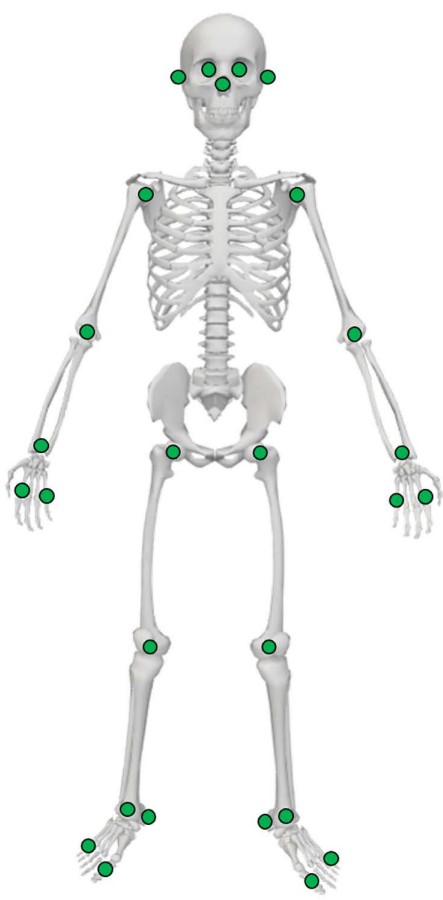

**Fig 3. Pose estimation keypoint locations.** Based on the COCO-Wholebody dataset [38].

trials. Rotational and translational KE of every segment (relative to CoM) were calculated and summed for total KE of each segment. Segment energies were then summed within limbs (bilateral legs, bilateral arms and trunk). CoM and limb energy timeseries were all cropped to a start point corresponding to the start of the serve (second local minimum prior to the most prominent peak (top of the serve) in the CoM vertical position data) and an end point corresponding to when the CoM position reached 8.5 m forward from the baseline (just before the player reached the net). The timeseries were split into two sections corresponding to the groundstroke sequence and the sprint, with a transition point defined as the change of direction (local minimum in resultant CoM velocity) as the player transitioned from the last groundstroke into the sprint. The increments and decrements across each energy timeseries were summed to give positive and negative work respectively for the CoM and limbs. Work done by CoM gave the external work and the sum of the work done by each limb gave the internal work. Total work done during a cycle of the protocol was taken as the groundstroke work multiplied by three, plus the sprint work.

To address the second aim, two different approximations of external work were also calculated using proxies for CoM. The first of these was based on a bounding box method, using the same Faster R-CNN [35] bounding boxes that were used for pose estimation (racket not included). The centres of 2D bounding boxes from each camera view were taken and the least squares intersection of the back-projected camera rays in 3D space was found. The trajectory of this point was filtered with a low-pass Butterworth filter (6 Hz cut-off frequency, determined by a residual analysis) and used as a

pseudo–CoM. The second CoM proxy was the pelvis segment origin of the OpenSim model. Whilst this still requires the full markerless pipeline, it has been used in this work to represent a generic fixed point on the player that could be tracked using other means, for example a mid-hip point. Mechanical work of these proxy points was calculated in the same way as for CoM.

## Statistical analysis

All statistical analysis was carried out in Python (Python 3.10). The peak forward velocity attained on each sprint (as a percentage of the player's overall maximum velocity) was correlated (Pearson's coefficient) against the mechanical work that the player had accumulated up until that point of the entire protocol. Pooled correlations, with 95% confidence intervals (CI), were calculated across all players, based on the method of Hopkins [41]. This method pools the individual correlations whilst accounting for random effects between groups and different sample sizes of each group. Correlation coefficients above 0.5, 0.7 and 0.9 are considered to be large, very large and extremely large respectively [42]. A sensitivity analysis ($\alpha=0.05$, power$=0.8$), performed using G*Power (Version 3.1.9.4), indicated that the minimum detectable effect with a sample size of 15 was $r=0.56$. Additionally, repeated measures Bland-Altman analyses [43] were used to compare the work done by the CoM and each of the proxies.

## Results

The players completed $6.3\pm2.2$ cycles of the fatiguing protocol. Peak CoM forward velocities in the baseline sprint test were $6.0\pm0.5$ m·s$^{-1}$ with mean reductions of $1.2\pm0.4$ m·s$^{-1}$ (~20%) across the protocol. Further results of the fatiguing protocol are given in Tables 1 and 2. The pooled correlation between peak sprint velocity (% of maximum sprint velocity) of each cycle and the cumulative positive work done was −0.93 (95% CI: −0.95, −0.88) for total mechanical work. External work only (neglecting internal work), as well as the bounding box and pelvis methods, all gave pooled correlations of −0.92 [−0.95, −0.88] (Fig 4). Results for negative mechanical work were almost identical so have been omitted for clarity.

A Bland-Altman plot comparing the work done by the CoM with each of the proxies is shown in Fig 5. The pelvis and bounding box methods yielded biases of 95 J·kg$^{-1}$ and 688 J·kg$^{-1}$ when compared to CoM, with LoA of ±58 J·kg$^{-1}$ and ±153 J·kg$^{-1}$ respectively. Additionally, an example of the CoM and proxy energies during one trial is shown in Fig 6.

## Discussion

This study aimed to assess whether mechanical work measured with our custom markerless motion capture system is an appropriate measure of workload that has on-court utility and to subsequently determine if approximations of mechanical work are sufficient. The total positive mechanical work done during the fatiguing protocol was strongly correlated with

**Table 1. Descriptive results of the fatiguing protocol, giving the number of cycles completed, peak velocity during the baseline sprint test, reduction in peak sprint velocity across the protocol and average and maximum heart rates across the whole protocol.**

| | | Number of cycles | Peak velocity (m·s⁻¹) | | Heart rate (bpm) | |
| --- | --- | --- | --- | --- | --- | --- |
| | | | Baseline | Change | Average | Maximum |
| Mean (± SD) | M | 6.0±2.4 | 6.3±0.4 | −1.3±0.4 | 175±5 | 189±5 |
| | F | 6.8±1.7 | 5.5±0.2 | −0.9±0.2 | 175±12 | 188±10 |
| Maximum | M | 12.0 | 7.0 | −2.2 | 180 | 195 |
| | F | 9.0 | 5.9 | −1.1 | 190 | 201 |
| Minimum | M | 4.0 | 5.5 | −0.9 | 168 | 179 |
| | F | 4.0 | 5.2 | −0.6 | 167 | 176 |

Results separated by males (M) and females (F).

**Table 2. Mean (± SD) of the positive mechanical work done during each cycle of the fatiguing protocol and the cumulative positive mechanical work across the entire protocol.**

| | $W_{tot}$ (J·kg$^{-1}$) | $W_{ext}$ (J·kg$^{-1}$) | $W_{int}$ (J·kg$^{-1}$) | $W_{pelvis}$ (J·kg$^{-1}$) | $W_{bb}$ (J·kg$^{-1}$) |
|---|---|---|---|---|---|
| Cycle | 351±60 | 234±43 | 117±19 | 330±64 | 923±163 |
| Cumulative | 2190±850 | 1453±555 | 737±301 | 2046±812 | 5740±2249 |

$W_{tot}$=Total mechanical work; $W_{ext}$=External mechanical work; $W_{int}$=Internal mechanical work; $W_{pelvis}$=External mechanical work from pelvis approximation; $W_{bb}$=External mechanical work from bounding box approximation.

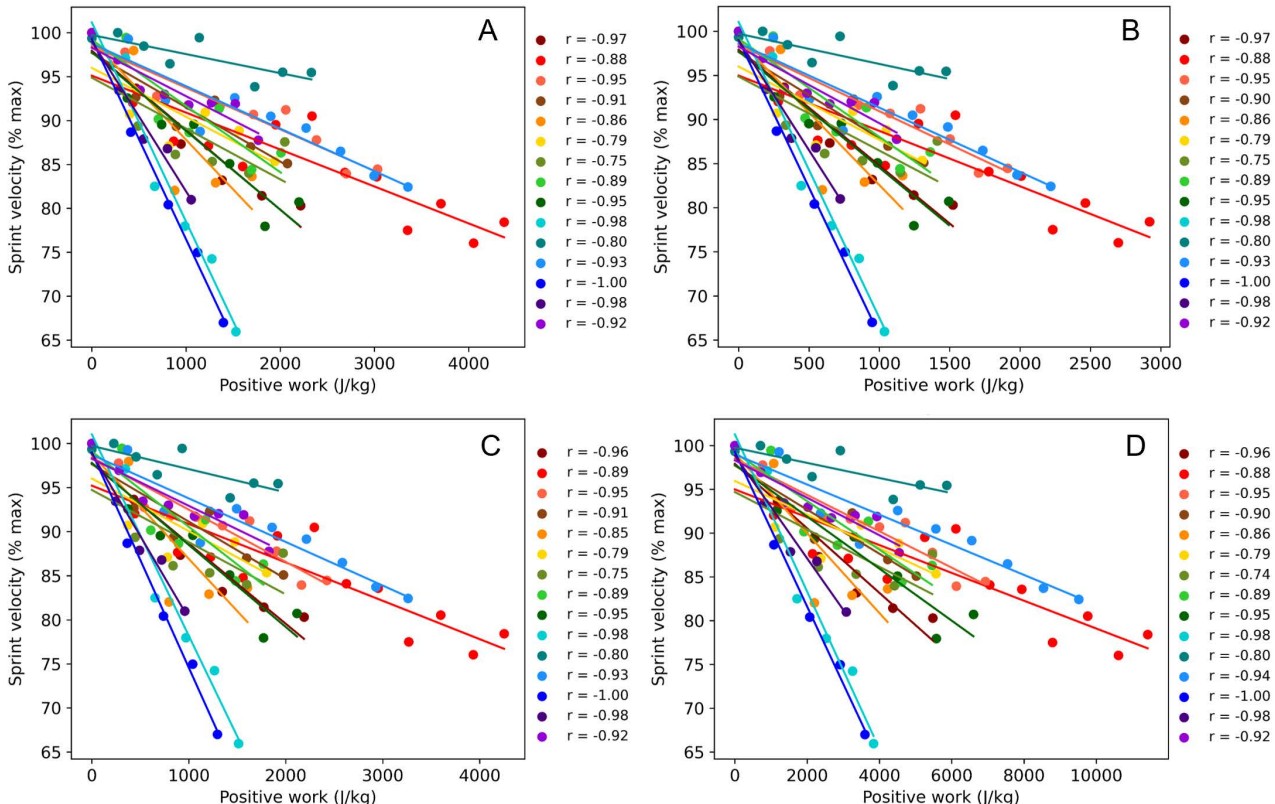

**Fig 4. Peak sprint velocity of each cycle against cumulative positive mechanical work for each player.** Each data point corresponds to a completed cycle of the protocol. Pearson's r coefficients given to the side of each plot. **(A)** Total mechanical work. **(B)** External work only. **(C)** Pelvis approximation. **(D)** Bounding box approximation.

reduction in sprint velocity, suggesting that this approach is indeed suitable for workload monitoring. Neglecting the internal work and using proxies for CoM had no clear effect on the correlations and so these simplifications could perhaps also be considered appropriate for the purpose of longitudinal monitoring of a single athlete in similar contexts (i.e., considering training or competition scenarios and court surfaces). However, the proxies both gave large systematic overestimates of the external work (Fig 5), which should be taken into account when deciding if these proxies are suitable. For example, the accuracy could be questionable when comparing across different players, court surfaces and types of play (competition or training).

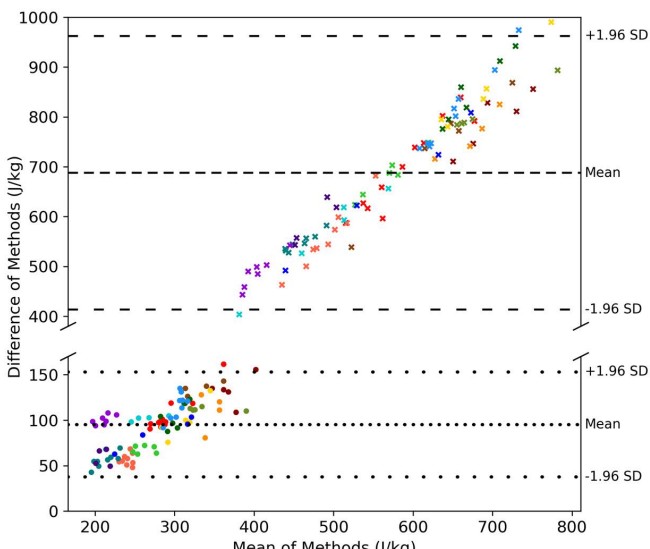

**Fig 5. Repeated measures Bland-Altman of positive external work, comparing the centre of mass (CoM) proxy methods (pelvis and bounding box) against the CoM work.** Each player is represented by a different colour. Pelvis method is represented by circles and dotted lines. Bounding box method is represented by crosses and dashed lines. (Note: For clarity, the scale of the y-axis is not uniform across the two sections of the figure.).

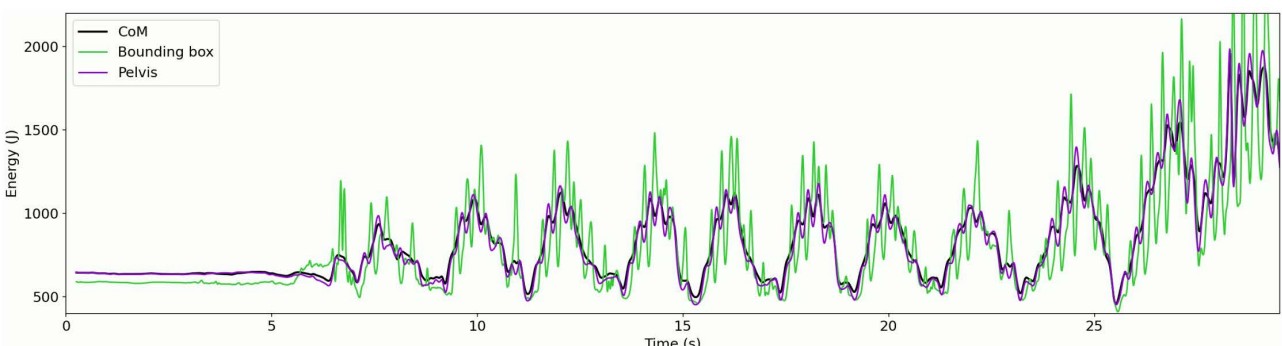

**Fig 6. Example of energies during one 30-s trial, for the centre of mass and proxy methods.**

The pooled correlation of −0.93 demonstrates the very strong association between mechanical work and acute neuro-muscular fatigue (reduction in maximum-effort sprint velocity), with the varying slopes reflecting the inter-individual variability in fatigue responses to the protocol (Fig 4). Even the weakest individual correlation (−0.75) can still be considered strong [42]. These results suggest that mechanical work is an appropriate indicator of fatigue and that our system could be implemented as a method for monitoring workload. Whilst this method may currently be more complicated and expensive compared to simpler wearable technologies, it has the key advantage of being completely non-invasive to players and allowing us to collect data that is ecologically valid, without interfering with a player's natural movement or comfort.

As a metric for external workload, mechanical work falls on the biomechanical side of the 'load' monitoring framework presented by Vanrenterghem and colleagues [44]. Biomechanical 'load' monitoring is often overlooked in favour of physiological measures, but the mechanical stresses on the musculoskeletal system have serious implications for adaptation and injury risk [44]. An average of 312 groundstrokes per match have been reported during male Grand Slam matches

[45] and each of these groundstrokes requires some degree of acceleration and deceleration. These movements are associated with high ground reaction forces that apply mechanical stresses to tissue, but are not fully captured by typical velocity or distance-related measures. Positive and negative external mechanical work are based on the changes in kinetic energy required to accelerate and decelerate the body CoM. Whilst this does not give a measure of the loading on tissues themselves, it does give an indirect indication of the cumulative accelerations and decelerations experienced by the musculoskeletal system. Therefore, we believe that mechanical work measured with markerless methods can provide more insight into external workload than other common metrics, such as distance or velocity-based measures.

Whilst mechanical work is our desired metric, calculating it from our markerless approach is computationally expensive, which limits the usability and accessibility of this type of tool. One simplification to our approach is to use only the external work. Internal work accounted for approximately a third of the total mechanical work in this study, so it cannot be neglected if an accurate measurement of mechanical work is desired. Indeed, Pavei and colleagues [46] have demonstrated the importance of internal work during the acceleration phase of sprinting, with it accounting for 41% of the total work. However, the almost identical relationship with reduction in sprint velocity for external work suggests that the external component is enough to provide an estimate of player fatigue and can therefore give an indication of workload. It should be noted that in our previous work [27], the internal work measured with our markerless system exhibited low systematic underestimation but high random errors (for serve, forehand and backhand movements) when compared against a *de facto* marker-based system. Although these random errors may balance out over multiple strokes, it is likely that there will be some errors in our values of internal work. We believe that, despite this underestimation, valuable information is still provided by computing internal work. However, if high precision, but not accuracy, of mechanical work is desired, then neglecting the internal work may be appropriate.

From a practical perspective, external work is more relatable to other commonly used metrics (focusing on whole-body movements) than total mechanical work; it indicates whole-body loading; and it is likely more understandable to players and coaches. Indeed, interviews with strength and conditioning coaches of elite tennis players (top 100 world ranking) have revealed the absence of established strategies for 'load' monitoring in tennis and a general lack of trust in validity of wearable technology [47]. It is possible that the inclusion of internal work may be seen to over-complicate workload monitoring. There has been recent interest in the use of external work as a workload metric in team sports where players are already using wearable tracking technology [25,48], making it simple for estimates of external work to be calculated from position and velocity data. However, the movement of these wearable devices is unlikely to accurately represent CoM movement. Indeed, segmental accelerometery-based methods have been demonstrated to overestimate mechanical loading variables compared to CoM loading [49]. Moreover, GPS and LPS devices are typically unable to detect within-stride energy fluctuations due to low sampling rates and aggressive filtering techniques [18,50], which would introduce further errors into calculations of work. There is real potential for the use of external mechanical work as a metric for quantifying workload in applied settings and video-based markerless approaches can offer a solution that overcomes limitations associated with wearable technology.

Using a fixed point as a proxy for CoM could reduce the computational cost of the data processing pipeline. The systematic increase in the between-method differences with the mean mechanical work (Fig 5) indicates proportional bias, reflecting systematic overestimation by the CoM proxy methods. These large systematic overestimates are a direct result of additional noise in the energy timeseries of the proxies in comparison to CoM (Fig 6). The bounding box method yielded work values over three times greater than those of CoM. This perhaps is not surprising as the dimensions of a bounding box can change rapidly if limbs are stretched out. Although this method greatly simplifies the data processing, it is not a good proxy for the CoM. This aligns with the findings of Javadiha and colleagues [51], who reported higher random errors for a bounding box method compared to pose estimation methods for estimating court position of padel tennis players. The overestimates from the pelvis method were less expected, as the pelvis segment and CoM were extracted from the model at the same stage. This demonstrates that a fixed point may not be a good approximation of CoM due to the

irregular movements performed during tennis and is consistent with the findings of Pavei and colleagues, who reported large overestimates of external mechanical work estimated from markers on the pelvis compared to a full-body marker set [52]. Despite the large errors seen for both CoM proxies, they appear to be systematic overestimates, with correlations with sprint velocity comparable between all methods. These approximations could be considered reasonable indicators of workload with the acknowledgement that the absolute estimates of external mechanical work are not an accurate representation of the work done by the CoM and that the metrics cannot be used interchangeably.

Using an anatomical point to represent CoM and estimate external work could potentially be a simpler solution for workload monitoring, and hence alternative computer vision techniques to the ones tested here should be explored. Object detection methods resulting in bounding boxes are unlikely to provide reasonable estimates of CoM, as demonstrated by our results. One option is to use the mid-point of 3D-reconstructed hip keypoints from pose estimation, which removes the need for the IK step. This method does not work well with our data, for which the hip keypoints were extremely noisy, highlighting the importance of the IK step in our pipeline and the current limitations of the 2D pose estimator. However, this should be possible with improved pose estimation models, trained on anatomically-accurate datasets. A different approach to reducing data processing requirements would be to reduce the amount of data itself, by decreasing the number of cameras or the frame rate. This should be explored by future research, with the possibility that this could eventually be achieved in real-time.

A limitation to this study is the capturing of only the third set of each cycle. Whilst the work done within each set is expected to be similar, any within-cycle variability is not captured, likely influencing the estimates of work. A further limitation is the repetitive nature of the protocol. During each cycle, players repeated the same drill and hence completed approximately the same amount of work each cycle. This was necessary due to data capture constraints of the markerless motion capture system, operating with eight cameras at full HD resolution (1920 x 1080) and a frame rate of 200 Hz. For each 30-s capture period, a minimum of two minutes was required for saving time, meaning it would be impossible to capture the whole protocol without unfeasibly long rest periods. This means that the capability of our approach to capture the fatigue response to the full range of movements observed in tennis is unclear. To comprehensively determine if the mechanical work approximations explored in this study are acceptable, this investigation needs to be repeated during open-play to allow for a greater variety of movements and intensities.

## Conclusion

Markerless motion capture can be implemented as a tool for monitoring on-court workload in tennis, with mechanical work providing a suitable metric that is very strongly associated with acute neuromuscular fatigue. The external work component alone provides an indication of workload and is perhaps more intuitive for practitioners and athletes than the internal component. Estimating external work with CoM proxies gave large systematic errors, but may also still provide reasonable indications of workload, which would greatly reduce data processing times and increase accessibility of this type of tool. However, it remains unclear if these associations would hold true in open-play. Therefore, we would recommend using the full mechanical work approach if practitioners are interested in the absolute values of total work done or comparing workload across different contexts (i.e., players, court surfaces and training or competition scenarios), but using just the external component if primarily interested in whole-body loading. Moreover, approximations of external work may be appropriate if just monitoring the training of an individual player and usability is more of a priority than accuracy. A tool of this nature would allow for non-invasive on-court workload monitoring in a range of settings, from training to competition.

## Acknowledgments

The authors sincerely thank Dr Murray Evans for his technical support with the markerless motion capture system, and Mr Zak Sheehy, Mr Jack Peters, Mr Filippo Santiano, Mr Seb Ison and Miss Ava Berglin for their assistance with data collection for this study.

## Author contributions

**Conceptualization:** Julie Emmerson, Laurie Needham, Sean Williams, Steffi Colyer.

**Data curation:** Julie Emmerson.

**Formal analysis:** Julie Emmerson.

**Funding acquisition:** Steffi Colyer.

**Investigation:** Julie Emmerson, Laurie Needham, Sean Williams, Steffi Colyer.

**Methodology:** Julie Emmerson, Laurie Needham, Sean Williams, Steffi Colyer.

**Project administration:** Julie Emmerson, Steffi Colyer.

**Resources:** Julie Emmerson.

**Software:** Julie Emmerson, Laurie Needham.

**Supervision:** Steffi Colyer.

**Validation:** Julie Emmerson.

**Visualization:** Julie Emmerson.

**Writing – original draft:** Julie Emmerson.

**Writing – review & editing:** Julie Emmerson, Laurie Needham, Sean Williams, Steffi Colyer.

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
