## [Decision Letter · Decision Letter 0]

29 Jan 2026

PONE-D-25-57000
Mechanical work derived using markerless motion capture provides a valid indication of acute neuromuscular fatigue in tennis
PLOS One

Dear Dr. Colyer,

Thank you for submitting your manuscript to PLOS ONE. After careful consideration, we feel that it has merit but does not fully meet PLOS ONE’s publication criteria as it currently stands. Therefore, we invite you to submit a revised version of the manuscript that addresses the points raised during the review process.

The points raised during the review process will serve to further strengthen your manuscript, and as this is a 'minor revision' addressing them should comfortably result in an accepted manuscript.

We look forward to receiving your revised manuscript.

Kind regards,

Daniel J. Glassbrook, PhD

Academic Editor

PLOS One

Journal Requirements:

4. Please expand the acronym “EPSRC” (as indicated in your financial disclosure) so that it states the name of your funders in full.

5. Please amend either the title on the online submission form (via Edit Submission) or the title in the manuscript so that they are identical.

6. Please amend your authorship list in your manuscript file to include author Steffi L Colyer.

7. Please amend the manuscript submission data (via Edit Submission) to include author Steffi Colye

8. Please include a caption for figure 3 and 4.

9. Please upload a copy of Figure 1, 2, to which you refer in your text on page 5, 6, 7, and 10. If the figure is no longer to be included as part of the submission please remove all reference to it within the text.

Reviewers' comments:

Reviewer's Responses to Questions

**Comments to the Author**

1. Is the manuscript technically sound, and do the data support the conclusions?

Reviewer #1: Yes

Reviewer #2: Yes

2. Has the statistical analysis been performed appropriately and rigorously?

Reviewer #1: Yes

Reviewer #2: Yes

3. Have the authors made all data underlying the findings in their manuscript fully available?

Reviewer #1: Yes

Reviewer #2: Yes

4. Is the manuscript presented in an intelligible fashion and written in standard English?

Reviewer #1: Yes

Reviewer #2: Yes

5. Review Comments to the Author

Reviewer #1: In this paper the external and internal mechanical work were computed during a tennis-specific on-court fatiguing task. The aim was to obtain a meaningful estimation of workload since, as indicated in the Introduction, commonly used metrics are non-validated or indirect (e.g. simple measures of frequency and duration of a training session).

As pointed out by the authors, wearable technology does not provide good accuracy for rapid movements such as those occurring in a tennis match and thus video-based tracking could be a valuable solution. The markerless system utilized in this study spares time, compared to a “standard” kinematic analysis and is more ecological than a lab setting. However, data analysis is still time demanding and a mocap system is quite expensive compared to a GPS system. Thus, I was expecting a comparison between these data and those that can be calculated when collected by means of a GPS system (only external work, in this case). Did the authors (or someone else) think about this?

Since in the literature it is common to estimate workload based on acute performance decrements, the authors hypothesized a correlation between mechanical work and “neuromuscular fatigue” (e.g. estimated by measuring decrements in speed).

Significant relationships were observed between these parameters either by considering total mechanical work, only external work and by calculating external work based on simpler datasets (as proxies of the CoM position).

I would suggest adding a figure/table with cumulative positive work data (mean and SD) of Wint, Wtot and W ext. The info reported in the discussion that Wint = 1/3 of W tot is very interesting and should be highlighted (but there are no values of W int in the paper). Then, I agree with the authors that computing only the external work will suffice if the aim is just to estimate changes in workload in a fatiguing task.

Specific points

Why the individual relationships in figure 1 have such different slopes? For some subjects the decrease in sprint velocity is greater than the increase in positive work, for others the contrary it is true. These differences are not discussed. Could you please speculate about this?

Bland Altman plots: there is a systematic increase in the difference of methods with an increase in the mean of methods, but this is not discussed. Could you please speculate about this?

Page 11, line 239: Neuromuscular fatigue is not maximum effort sprint velocity but the difference in velocity pre-post the fatiguing task; please check the entire manuscript and amend.

page 11, line 240-242: This statement “These results suggest that mechanical work is an appropriate indicator of fatigue and that our system could be implemented as a method for monitoring workload” does not consider that this method is not easy to use and quite expensive in comparison to others, even if more ecological (it could not be implemented easily, as with GPS technology).

Page 12, line 245-248: In these lines you state: “As a metric for external workload, mechanical work gives a reflection of the mechanical energies involved and hence could be argued to fall on the physiological side of the ‘load’ monitoring framework presented by Vanrenterghem and colleagues, but it can also give some insight into the biomechanical ‘load’.”

I do not really follow this line of reasoning: mechanical work calculations are the object of biomechanical studies. Investigating the physiological side of load monitoring would imply measures of metabolic energy expenditure; eventually phjysiological/metabolic data could be estimated from mechanical data by knowing/assuming a value of efficiency (as in Osgnach et al. 2024).

Page 12, lines 257-259: I do also believe that mechanical work measured with markerless methods can provide more insight into external workload than other common metrics, such as distance or velocity-based measures. This is indeed an interesting and useful paper.

Page 12, lines 262-264: This is an interesting info (Wint = 1/3 of W tot). Please provide cumulative positive work data (mean and SD) of Wint, Wtot and W ext (see comment above).

Pages 13-14: In the discussion regarding the validity of approximations of the COM position in the determination of external mechanical work you could also consider the paper of Pavei et al. (2017) who investigated the same matters in walking and running

Pavei et al. (2017) On the estimation accuracy of the 3D body center of mass trajectory during human locomotion: inverse vs. forward dynamics. Front Physiol https://doi.org/10.3389/fphys.2017.00129

Reviewer #2: Thank you for your submission. This paper describes a study that examined the potential application of mechanical work derived with markerless (ML) motion capture as an indicator of acute fatigue in tennis players. The reported findings are clinically relevant as they could address an applied problem of non-invasively monitoring workload in athletic movements. The paper is well-written and appears to meet the journal's requirements of data availability, human subjects protection, and reporting. The following annotations will hopefully provide the authors with possible areas of improvement in clarity and rigor:

GENERAL

While the tennis racket minimally influences external work, would the bounding-box proxy be more sensitive to racket motion, especially considering the authors reported larger systematic overestimates with this COM proxy?

Lines 72-75: Mechanical work and more broadly, energy flow, have certainly been used to assess the efficiency of gait and sports-specific movements. However, the distinction between mechanical work and more common distance- or velocity-based workload metrics may not be immediately intuitive to a general readership. Clarify why mechanical work is conceptually different from and potentially more informative than distance- or speed-based metrics in capturing cumulative accelerations and deceleration.

Lines 130-134: The authors captured only the third set due to storage limitations. This is understandable and it is reasonable to assume that the work done in the first two sets is "sufficiently similar to that of the captured third set." However, the authors should explicitly acknowledge, perhaps in the Limitations section, this as a modeling assumption and briefly note that unmeasured within-cycle variability (e.g., fatigue-related changes across sets) could influence absolute work estimates, even if relative trends are expected to remain consistent.

Lines 193-194: Post-hoc power analyses are generally considered redundant once effect sizes and confidence intervals are reported. The authors may wish to explicitly frame it as a descriptive sensitivity analysis included solely to contextualize sample size, rather than as inferential support.

Line 314: The authors describe the hip midpoint as "extremely noisy." It would be helpful to clarify whether this instability reflects noise amplification from combining left and right hip estimates, rather than implying that the individual hip joint centers themselves are extremely noisy. This distinction would help interpreting the limitations of midpoint-based COM proxies more accurately.

6. PLOS authors have the option to publish the peer review history of their article (what does this mean?). If published, this will include your full peer review and any attached files.

Reviewer #1: No

Reviewer #2: No

---

## [Author Response · Author response to Decision Letter 1]

11 Mar 2026

Journal Requirements:

Style requirements have been checked.

Data availability statement has been checked.

This has been corrected.

4. Please expand the acronym “EPSRC” (as indicated in your financial disclosure) so that it states the name of your funders in full.

Funding information has been included in the cover letter, with funder names in full.

5. Please amend either the title on the online submission form (via Edit Submission) or the title in the manuscript so that they are identical.

This has been corrected.

6. Please amend your authorship list in your manuscript file to include author Steffi L Colyer.

Done.

7. Please amend the manuscript submission data (via Edit Submission) to include author Steffi Colyer

“Steffi Colyer” changed to “Steffi L Colyer” in the manuscript. This is now consistent with the author list on the manuscript submission data.

8. Please include a caption for figure 3 and 4.

Captions for figures 3 and 4 are on lines 158 and 207 respectively.

9. Please upload a copy of Figure 1, 2, to which you refer in your text on page 5, 6, 7, and 10. If the figure is no longer to be included as part of the submission please remove all reference to it within the text.

These figures have been uploaded.

We have considered the suggestion of reviewer 1 to cite a specific study, and this has now been included.

Reference list has been checked.

We would like to thank the reviewers for taking the time to read our paper and providing thoughtful and constructive feedback.

Reviewer #1:

General

1. As pointed out by the authors, wearable technology does not provide good accuracy for rapid movements such as those occurring in a tennis match and thus video-based tracking could be a valuable solution. The markerless system utilized in this study spares time, compared to a “standard” kinematic analysis and is more ecological than a lab setting. However, data analysis is still time demanding and a mocap system is quite expensive compared to a GPS system. Thus, I was expecting a comparison between these data and those that can be calculated when collected by means of a GPS system (only external work, in this case). Did the authors (or someone else) think about this?

We agree that markerless mocap is expensive and time-demanding (comment added on line 251-2), and that it would have been interesting to include a comparison to mechanical work derived from a wearable system, such as GPS (or LPS for indoor activity). However, we did not have access to such systems. Further work should look to compare these results (external work only) with wearables.

2. I would suggest adding a figure/table with cumulative positive work data (mean and SD) of Wint, Wtot and W ext. The info reported in the discussion that Wint = 1/3 of W tot is very interesting and should be highlighted (but there are no values of W int in the paper). Then, I agree with the authors that computing only the external work will suffice if the aim is just to estimate changes in workload in a fatiguing task.

Thank you for this suggestion. A table has been added with these values (Table 2).

Specific

1. Why the individual relationships in figure 1 have such different slopes? For some subjects the decrease in sprint velocity is greater than the increase in positive work, for others the contrary it is true. These differences are not discussed. Could you please speculate about this?

This is an important observation. The individuals demonstrated varying fatigue responses to the protocol, with some players fatiguing quicker than others (as would be expected). A comment has been added (lines 247-8) to explain this.

2. Bland Altman plots: there is a systematic increase in the difference of methods with an increase in the mean of methods, but this is not discussed. Could you please speculate about this?

A sentence has been added (lines 302-3) to explain this systematic increase. It likely reflects the proportional bias arising from a consistent overestimate by the CoM proxy methods.

3. Page 11, line 239: Neuromuscular fatigue is not maximum effort sprint velocity but the difference in velocity pre-post the fatiguing task; please check the entire manuscript and amend.

Thank you for pointing this out. We have checked for this mistake throughout the manuscript and amended the wording where appropriate (line 247).

4. page 11, line 240-242: This statement “These results suggest that mechanical work is an appropriate indicator of fatigue and that our system could be implemented as a method for monitoring workload” does not consider that this method is not easy to use and quite expensive in comparison to others, even if more ecological (it could not be implemented easily, as with GPS technology).

We agree that there are still challenges of this method. We have added a comment (line 251-2) to acknowledge the complexity and cost of this method.

5. Page 12, line 245-248: In these lines you state: “As a metric for external workload, mechanical work gives a reflection of the mechanical energies involved and hence could be argued to fall on the physiological side of the ‘load’ monitoring framework presented by Vanrenterghem and colleagues, but it can also give some insight into the biomechanical ‘load’.” I do not really follow this line of reasoning: mechanical work calculations are the object of biomechanical studies. Investigating the physiological side of load monitoring would imply measures of metabolic energy expenditure; eventually physiological/metabolic data could be estimated from mechanical data by knowing/assuming a value of efficiency (as in Osgnach et al. 2024).

We have carefully considered this point, and we agree that it may be slightly misleading. Mechanical work (in this context) has been derived from velocities which are on the physiological side, so it could be argued this way, but primarily it is a biomechanical metric. This sentence has been rewritten to clearly state that mechanical work is on the biomechanical side of the framework (lines 255-7).

6. Page 12, lines 257-259: I do also believe that mechanical work measured with markerless methods can provide more insight into external workload than other common metrics, such as distance or velocity-based measures. This is indeed an interesting and useful paper.

Thank you for this feedback.

7. Page 12, lines 262-264: This is an interesting info (Wint = 1/3 of W tot). Please provide cumulative positive work data (mean and SD) of Wint, Wtot and W ext (see comment above).

A table has been added with these values.

8. Pages 13-14: In the discussion regarding the validity of approximations of the COM position in the determination of external mechanical work you could also consider the paper of Pavei et al. (2017) who investigated the same matters in walking and running

This is a good suggestion. We have now referred to this paper on lines 313-5.

Reviewer #2:

1. While the tennis racket minimally influences external work, would the bounding-box proxy be more sensitive to racket motion, especially considering the authors reported larger systematic overestimates with this COM proxy?

The bounding box only contains the person, and not the tennis racket. This has now been clarified on line 177.

2. Lines 72-75: Mechanical work and more broadly, energy flow, have certainly been used to assess the efficiency of gait and sports-specific movements. However, the distinction between mechanical work and more common distance- or velocity-based workload metrics may not be immediately intuitive to a general readership. Clarify why mechanical work is conceptually different from and potentially more informative than distance- or speed-based metrics in capturing cumulative accelerations and deceleration.

Thank you for pointing out that this wasn’t clear. We have added a sentence (lines 74-6) to explain this.

3. Lines 130-134: The authors captured only the third set due to storage limitations. This is understandable and it is reasonable to assume that the work done in the first two sets is "sufficiently similar to that of the captured third set." However, the authors should explicitly acknowledge, perhaps in the Limitations section, this as a modeling assumption and briefly note that unmeasured within-cycle variability (e.g., fatigue-related changes across sets) could influence absolute work estimates, even if relative trends are expected to remain consistent.

We agree that this limitation should be explicitly stated. A sentence has been added to the discussion on lines 333-5.

4. Lines 193-194: Post-hoc power analyses are generally considered redundant once effect sizes and confidence intervals are reported. The authors may wish to explicitly frame it as a descriptive sensitivity analysis included solely to contextualize sample size, rather than as inferential support.

This sentence has been slightly reworded. The word ‘power’ has been removed from ‘sensitivity power analysis’ to suggest that it is descriptive, rather than providing inferential support.

5. Line 314: The authors describe the hip midpoint as "extremely noisy." It would be helpful to clarify whether this instability reflects noise amplification from combining left and right hip estimates, rather than implying that the individual hip joint centers themselves are extremely noisy. This distinction would help interpreting the limitations of midpoint-based COM proxies more accurately.

This sentence (line 326) has been reworded to clarify that both hip keypoints were very noisy, prior to combining them for the midpoint estimation.

(Line numbers refer to the manuscript with track changes)

---

## [Editor Report · Decision Letter 1]

13 Mar 2026

Mechanical work derived using markerless motion capture provides a valid indication of acute neuromuscular fatigue in tennis

PONE-D-25-57000R1

Dear Dr. Colyer,

We’re pleased to inform you that your manuscript has been judged scientifically suitable for publication and will be formally accepted for publication once it meets all outstanding technical requirements.

Kind regards,

Daniel J. Glassbrook, PhD

Academic Editor

PLOS One
---

## [Editor Report · Acceptance letter]

PONE-D-25-57000R1

PLOS One

Dear Dr. Colyer,

I'm pleased to inform you that your manuscript has been deemed suitable for publication in PLOS One. Congratulations! Your manuscript is now being handed over to our production team.

Kind regards,

on behalf of

Dr. Daniel J. Glassbrook

Academic Editor

PLOS One